# Changes in the Texture and Flavor of Lotus Root after Different Cooking Methods

**DOI:** 10.3390/foods12102012

**Published:** 2023-05-16

**Authors:** Chunlu Qian, Yaying Jiang, Yan Sun, Xiaodong Yin, Man Zhang, Juan Kan, Jun Liu, Lixia Xiao, Changhai Jin, Xiaohua Qi, Wenfei Yang

**Affiliations:** 1Department of Food Science and Engineering, School of Food Science and Engineering, Yangzhou University, Yangzhou 225127, China; 2Huaiyin Institute of Agricultural Sciences in Xuhuai Area of Jiangsu, Huaian 223001, China; 3Department of Horticulture, College of Horticulture and Landscape Architecture, Yangzhou University, Yangzhou 225009, China

**Keywords:** lotus root, cooking, texture, free amino acids, 5′-Nucleotides, volatiles

## Abstract

The changes in the texture and flavor of lotus root were determined before and after boiling, steaming and frying. Compared to fresh lotus root, all three kinds of cooking decreased the hardness and springiness, and frying significantly enhanced the gumminess, chewiness and cohesiveness. The flavor components, such as flavor amino acids, nucleotides and their taste character in lotus roots, were determined by liquid chromatography and electronic tongue. The amino acids and nucleotide contents of fresh lotus root were 20.9 and 0.07 μg/kg, respectively. The content of flavor substances in lotus roots decreased obviously, and the texture characteristics decreased after boiling and steaming. After deep-frying for 2 min, the free amino acids and nucleotide contents of lotus root were 32.09 and 0.85 μg/kg, respectively, which were the highest in all cooking methods. The contents of volatile flavor components and their smell character in lotus roots were determined by GC-MS and electronic nose. There were 58 kinds of flavor compounds identified in fresh lotus root, mainly alcohols, esters and olefins. The total amount of volatile flavor compounds decreased, and new compounds, such as benzene derivatives, were produced in lotus roots after boiling and steaming. After deep-frying, the content of volatile flavor compounds in lotus root increased significantly, especially the aldehyde volatile flavor compounds. The production of pyran, pyrazine and pyridine volatile flavor compounds made the lotus root flavor unique and delicious. The taste and smell character of lotus roots before and after cooking were effectively separated by an electronic tongue, nose and PCA analysis; the results suggested the boiled lotus root exhibited the most natural and characteristic taste and smell among the four groups.

## 1. Introduction

Lotus (*Nelumbo nucifera* Gaertn), which belongs to one of two species of water plants in the Lotus family, is a perennial aquatic plant that is widely cultivated as a vegetable throughout the East. Lotus has great economic value because its root exhibits a desirable crisp texture and white color, is rich in nutrients and is widely consumed all over the world. As a popular vegetable, lotus root can be eaten in roasted, pickled, dry-sliced and fried forms, exhibiting a variety of nutritional properties [1]. Fresh and cooked Lotus roots are rich in beneficial substances like flavonols, alkaloids, lipids, phospholipids, flavonoids, carotenes, lutein and many minerals [2]. In China, Lotus root is cultivated all over the country, especially in provinces with abundant wetlands, such as Jiangsu and Hubei. Lotus root could be used as a major or minor ingredient to make traditional foods, including soups, cold dishes, fried clips and baked desserts. Although there is considerable interest in determining the bioactive compounds and factors of lotus root that influence its processing and cooking quality, there is still no report on its texture and flavor characteristics in raw and cooked material.

This study aimed to determine the changes in the texture and flavor of raw lotus root and materials after different cooking methods, such as boiling, steaming and frying for different durations. These three heating methods were the most common processing way of lotus root and derived a series of processing products, so entrepreneurs, consumers and researchers paid great attention to the difference between these heating methods on the quality of lotus root products. In this study, the non-volatile and volatile flavor components were identified by high-performance liquid chromatography and headspace solid-phase micro-extraction gas chromatography (HS-SPME-GC-MS), and their flavor character was estimated by electronic tongue and nose. These equipment and methods were effectively applied to analyze the flavor of coffee, meat and so on [3,4]. The information provided here contributes to the understanding of the physicochemical and flavor properties of lotus root that may affect the texture and taste, as well as their changes after the process. In addition, this research forms the basis for further research on improving the quality of processed lotus root products.

## 2. Materials and Methods

### 2.1. Raw Materials

Fresh lotus roots (*Nelumbo nucifera* G. cv. ‘Zhenzhu’) were collected from Aquatic Vegetable Experimental Base of Yangzhou University in Jiangsu Province and transported to laboratory within 1 h. The lotus roots used are grown using standard cultivation methods. After washing and drying, lotus roots with similar size, consistent maturity and no obvious damage or microbial infection were selected as test materials; each lotus root joint was about 1 kg. After removing the skin manually with a sharp knife, the edible portion of lotus roots was cut into slices about 0.2 cm thick.

### 2.2. Cooking Procedures for Lotus Root

Raw lotus root slices with similar size and shape were selected for the following treatments: (1) raw lotus root slices in control group, without any treatment; (2) raw lotus root slices were boiled in boiling water at 100 °C for 2 min, 4 min and 6 min; (3) raw lotus root slices were steamed at 100 °C in a steaming pan for 2 min, 4 min and 6 min; and (4) raw lotus root slices were fried at 165 °C in a deep-frying pan for 1 min, 2 min and 3 min, respectively. At least three random duplicates of each sample were selected for experiment.

### 2.3. Determination of Physicochemical and Textural Properties of Lotus Root

The water content in lotus root was determined by putting the raw lotus root in a drying box at 105 °C and weighing it after drying for 24 h to calculate the weight difference between the dried and original sample. The percentage of weight difference in fresh lotus root was the water content [5]. The determination of soluble solid (SSC, in °Brix) in lotus root was made by grinding the lotus root in a mortar and dropping a few drops of juice on the refractive prism of the refractometer (PR32; tago Co., Ltd., Tokyo, Japan). For each measurement, five lotus roots were used. The soluble sugar content and titratable acid content in fresh lotus root were measured accordingly [6], and each sample was repeated in triplicate.

The texture profile analysis (TPA) used a texture analyzer equipped with TMS-Pro (Food Technology Corporation, Sterling, VA, USA), in which Texture Lab Pro (texture Index 32 software) is equipped with a cylindrical probe (P/45). The mass meter is programmed to move the cylindrical flat-end punch (4.5 cm in diameter) downward from 16 mm above the sample surface. The test speed is 60.00 mm/s. The lotus root sample (0.6 cm) was placed on a flat aluminum base, repeated 9 times, with a compression ratio of 70%, kept for 5 s, and then compressed again [7]. The TPA was conducted on the cross-section of lotus root slices.

### 2.4. Chemicals and Regents

Standards of amino acids, Ala, Arg, Asp, Gly, Glu, Ile, Leu, Lys, His, Met, Phe, Ser, Val, Thr, Trp and Tyr, and Nucleic acid, 5′-CMP, 5′-UMP, 5′-GMP, 5′-IMP and 5′-AMP, were purchased from Yuanye Biotechnology Co., Ltd. (Shanghai, China). Hydrochloric acid, boric acid, sodium dihydrogen phosphate, disodium hydrogen phosphate, b-mercaptoethanol, glacial acetic acid and tetrabutylammonium hydroxide, and HPLC grade of acetonitrile, methanol, were purchased from Sinopharmaceutical Group Chemical Co., Ltd. (Shanghai, China). The Inertsil-ODS-SP-C18 cartridge was purchased from Shimadzu Co., Ltd. (Shinjuku-ku, Japan).

### 2.5. Determination of Free Amino Acids

The preparation process of lotus root sample is as follows: 1 g lotus root sample was ground with distilled water and extracted at 80 °C for 20 min. After cooling to room temperature, the extract was prepared with ultra-pure water to 10 mL and filtered with 0.45 μm nylon membrane. The inertsil-ODS-SP-C18 column (Shimadzu Co., Ltd., Shinjuku-ku, Japan) was activated with 100% methanol at 1 mL/min flow rate for 30 min. A total of 70 μL of lotus root extract or standard amino acid solution was mixed with 10 μL o-phthalaldehyde (OPA) solution and incubated at 25 ± 1 °C for 2 min. The reaction mixture was then immediately used for HPLC analysis. Before pre-column derivatization with OPA, it was filtered by 0.45 μm nylon membrane. The derivation of OPA was configured as follows: 5 mg of OPA was dissolved in methanol (0.05 mL), mixed with 0.45 mL boric acid (0.4 M, PH 7.5), then 25 μL of β-mercaptoethanol was added [8].

Amino acids were determined by Waters E2695 series HPLC (Waters Technologies, Milford, MA, USA) and separated on inertsil-ODS-SP-C18 column (250 × 4.6 mm, Shimadzu). The column temperature was 40 °C. The mobile phase was methanol/acetonitrile/water (45/45/10, A) and phosphate buffer (pH 7.5, B). The mobile phase was filtered through a 0.22 µm membrane filter and degassed prior to use. The elution used the linear gradient shown in Appendix A. The flow rate was 1.0 mL/min. The detection wavelength of DAD was 338 nm, and the derivative amino acids were detected. The injection volume was 20 μL.

### 2.6. Determination of 5′-Nucleotide

Additionally, 1 g of lotus root samples were ground with distilled water (10 mL) and extracted at 100 °C for 5 min. After cooling to room temperature, the extract was centrifuged at 4500× *g* for 10 min. The supernatant was prepared with ultrapure water to 10 mL and filtered with 0.45 μm nylon membrane [9]. The analysis system of nucleotides is the same as that of free amino acids. The chromatographic column was Inertsil-ODS-SP-C18 column (250 × 4.6 mm, Shimadzu); the mobile phase was distilled water–methanol–glacial acetic acid–tetrabutylammonium hydroxide (894.5 × 100 × 5 g, *v*/*v*); the injection volume was 20 μL; the flow rate was 0.7 mL/min; and the detection wavelength was 254 nm.

### 2.7. Determination of Taste Trait by Electronic Tongue

Electronic tongue (Shanghai Bosin Industrial Development Co., Ltd., Shanghai, China) was used to analyze the changes in lotus root flavor after cooking treatment. The system consists of automatic sampler and sensor array: CA0 (detection of sour substances), C00 (detection of bitter substances), AE1 (detection of bitter substances), AAE (detection of fresh substances) and CT0 (detection of salt substances). It is composed of reference electrodes (Ag/AgCl) and data analysis software BOSIN. The electronic tongue contains five chemical sensors with organic film coatings, each of them with specific sensitivity and selectivity. Before the experiment, the E tongue sensor was activated with 0.01 M KCl solution to stabilize the measurement signal. Then, the sensor was cleaned with deionized water for 10 s. In the aspect of sample preparation, we used the method described and made some modifications [10,11]. The 5 g lotus root sample was mixed in 50 mL of 55 °C distilled water, stirred for 10 min, and then centrifuged at 3000× *g* for 10 min. After centrifugation, the supernatant was filtered and prepared for analysis with electronic tongue. Each sample was measured in triplicate, and the average value was taken for further analysis.

### 2.8. Determination of Smell Characteristics by Electronic Nose

Electronic nose analysis was conducted (Shanghai Bosin Industrial Development Co., Ltd.). The electronic nose consists of (1) a sampling system and a gas acquisition system, (2) a sensor array and (3) an intelligent recognition system. The electronic nose consists of 18 metal oxide sensors, each of which is sensitive to different volatile compounds. Table 1 lists the main uses of 18 sensors. According to the method with a slight change, 5 g lotus root samples were put into a sample bottle and incubated at 55 °C for 20 min. The test parameters were as follows: airflow (1 L min^−1^), test time (240 s) and cleaning time (120 s) [12]. Each sample was measured in triplicate; the average value was taken for further analysis.

### 2.9. HS-SPME-GC-MS Analysis

The HS-SPME-GC-MS analysis of lotus roots was performed as described [3]. Before analysis, 5 ± 0.0005 g of each lotus root sample was cut up and put into a 20 mL glass bottle containing 3 mL sodium chloride (saturation) and 10 μL 1pyr2-dichlorobenzene as the internal standard. The bottle is then sealed with a polytetrafluoroethylene-silica gel diaphragm and mixed by magnetic stirring. Each analysis was carried out in duplicate with different vials.

Under the condition of headspace (HS), the fiber used to extract volatile components was polydimethylsiloxane (PDMS) 65 μm. Before analysis, the fiber was preheated at 250 °C for 20 min. The fiber was inserted into the sample bottle through the diaphragm and exposed to HS at 55 °C for 30 min to collect analytes. The distance between the fiber tip and the sample bed was about 1 cm. Then, the fiber was taken out of the vial and inserted into the injection port of the GC-MS instrument.

GC conditions: The analysis of volatile compounds was performed on GC-MS apparatus (Trace TSQ, Thermo Fisher Scientific, Waltham, MA, USA). The analyte’s removal from the fiber was carried out by holding the injector temperature at 250 °C. Volatiles were separated using DB-5MS (33 m × 250 mm I.D., film thickness 0.25 μm) column, carrier gas He, flow rate 0.8 mL/min, split ratio 10:1, injection temperature 250 °C, column temperature program: initial temperature 40 °C, 5 °C/min increased to 65 °C, kept 3 min, 5 °C/min to 150 °C, kept 4 min, 10 °C/min to 210 °C, kept 2 min. Thermal desorption for 7 min was carried out at 250 °C [13].

The MS conditions were as follows: electron ionization source temperature was maintained at 250 °C; ionization mode, EI; transmission line temperature, 250 °C; electron energy, 70 eV; detector voltage, 350 V; and mass sweep range 33–300 amu.

### 2.10. Statistical Analyses

Completely random design was adopted throughout the research process. The experiment was repeated in triplicate. The data were statistically analyzed by Excel1 2.0 software and expressed as mean ± standard deviation (mean ± SD). Principal component analysis (PCA) was conducted by Rstuido, radar fingerprint analysis was conducted by Origin, and variance (ANOVA) analysis was carried out by SPSS Statistics 26.

## 3. Result

### 3.1. Physicochemical and Textural Properties of Fresh Lotus Roots

The water content of the raw lotus root was 82.99%; the soluble solids constituted 8.03%; the soluble sugar content was 27.02 mg/g; and the titratable acid content was 0.32% (Table 2).

### 3.2. Texture Properties of Lotus Roots before and after Cooking

Lotus root is famous for its hard and brittle texture. The hardness and springiness of lotus roots decreased after all three kinds of cooking, but the hardness and springiness then increased significantly (*p* < 0.05) after frying for 2 min (Table 3). Lotus root kept high springiness after frying for 3 min, whereas its hardness declined significantly (*p* < 0.05). The gumminess, chewiness and cohesiveness of lotus root were increased significantly (*p* < 0.01) after frying, and a decline showed after frying for 3 min. The boiling and steaming did not affect gumminess, chewiness and cohesiveness significantly (*p* > 0.05) (Table 3).

### 3.3. Free Amino Acid Contents of Lotus Roots before and after Cooking

Free amino acid is the basic unit of protein and can also make food have a special flavor. The total amino acid content of fresh lotus root was 20.90 mg/g, and the highest content of single amino acid was threonine, 10.82 mg/g (Table 4). The bitter amino acid content was 7.40 mg/g, accounting for 35.41% of the total amino acid content. The sweet amino acid content was 11.94 mg/g, accounting for 57.13% of the total content. After boiling and steaming, most amino acid content in lotus roots decreased obviously, and the longer the cooking time, the lower the amino acid content. Some amino acid content in lotus root increased after frying, and MSG-like amino acid content increased because the new flavor of amino acid-glutamic acid was produced (Table 4).

### 3.4. 5′-Nucleotides Contents of Lotus Roots before and after Cooking

The 5′-nucleotides are important flavor substances. As shown in Table 5, fresh lotus root contained 0.07 mg/g of 5′-nucleotides, just had 5′-CMP and 5′-UMP, and their contents were 0.05 and 0.02 mg/g, respectively. After cooking, the 5′-nucleotides content in lotus roots increased at first but decreased after cooking for a long time. The 5′-GMP was produced in lotus roots after boiling and steaming, and abundant 5′-IMP was produced in the lotus root after frying for a long time.

### 3.5. Electronic Tongue for Taste Character

The sweet, fresh, bitter, salty and sour tastes of lotus roots were collected by the electronic tongue, and the taste changes in fresh and cooked lotus roots were measured. As shown in Figure 1, fresh lotus root had a highly bitter, salty and sweet taste. The response values of the three categories sweet, bitterness and umami of lotus roots decreased gradually with the extension of boiling and steaming time, and the response values of salty and sour declined with no significant (*p* > 0.05) difference between cooking times, with the exception that salty response value of boiled lotus root increased. The response values of the bitterness, sweet and salty categories of deeply fried lotus roots decreased sharply, while its fresh response value increased significantly (*p* < 0.01).

The flavors of lotus roots before and after cooking were effectively separated by principal component analysis. As shown in Figure 2, the contribution rate of the PC1 was 93.47%, the contribution rate of the PC2 was 0.19% and the total was 93.66%, indicating that the two-dimensional scatter diagram of principal component analysis could reflect the taste differences of lotus roots after different cooking methods. The four groups of lotus root samples were clearly separated from each other on PC1, and there was no overlap, indicating that each group of samples had relatively different taste characteristics. -

### 3.6. Electronic Nose for Smell Character

The electronic nose, which was equipped with 18 kinds of sensors (Table 1), was used to analyze the volatile flavor compounds of lotus roots before and after different cooking methods. As shown in Figure 3, the signal values of 18 sensors varied with cooking methods and cooking time. The odors of lotus roots from different cooking methods were effectively separated by principal component analysis. As shown in Figure 4, the contribution rate of the PC1 was 88.70%, the contribution rate of the PC2 was 1.63% and the total was 90.33%, indicating that the two-dimensional scatter diagram of principal component analysis could reflect the odor differences of lotus roots before and after different cooking methods. The four groups of lotus roots were clearly separated from each other on PC1 and PC2, except some overlapping areas that appeared between boiled and steamed lotus roots, which indicated that each group of samples had relatively different odor characteristics and there were similarities in the smell between boiled and steamed lotus roots.

### 3.7. Volatile Aroma Components Contents of Lotus Roots before and after Cooking

The volatile flavor compounds in lotus roots before and after different cooking methods were determined by HS-PME-GC-MS. According to the changes in the texture and taste of lotus roots after cooking, the lotus roots with better texture were selected and boiled and steamed for 4 min and fried for 2 min. As shown in Table 6, the volatile flavor compounds in fresh lotus root changed significantly after different cooking methods. A total of 58 kinds of volatile flavor compounds were identified in fresh lotus root, and the total content was 427.5 μg/Kg. The contents and types of volatile flavor compounds in fresh lotus root decreased after boiling and steaming. A total of 49 and 56 kinds of volatile flavor compounds were identified in lotus roots after boiling and steaming, and the total contents were 277.88 and 365.14 μg/Kg, respectively. A total of 38 kinds of volatile flavor compounds in lotus root were identified after deep-frying, but the total content increased by 8.73 times, and a large number of aldehydes and pyran compounds were produced.

## 4. Discussion

The water content of raw Zhenzhu lotus root was similar to former reports on lotus root [14] and potato [15,16], indicating that the aquatic nature of lotus root did not result in higher water content than other tuber plants. The content of soluble solids in Zhenzhu lotus root was consistent with a report about lotus root [17]. The contents of soluble sugar and titratable acid in lotus roots were higher than those of other reported lotus roots (Table 2) [18,19]. The hardness of fresh lotus root is similar to that reported of potato, but its elasticity, stickiness and cohesion are significantly higher than those of potato (Table 3) [20]. In general, strong cohesion makes the lotus root more capable of withstanding the pressure of packaging and transportation.

The texture of lotus root became soft, and the hardness and elasticity decreased after the cooking treatment. After boiling, steaming and frying, the hardness of lotus roots decreased by up to 73.71%, 69.82% and 25.44%, respectively, and the springiness decreased by up to 71.43%, 66.84% and 33.16%, respectively (Table 3). This is mainly due to cooking heat treatment resulting in the decomposition of intercellular mucus, weakening of the cell wall, softening of the texture and reduction of the brittleness in plant tissue [19]. After boiling, steaming and frying, the gumminess of lotus roots increased at first and then decreased (Table 3). The increase in the gumminess of lotus roots after cooking may be due to the gelatinization of starch [21]. The chewiness of lotus root decreased after boiling but increased and reached the maximum after steaming for 4 min. After deep-frying, the chewiness of lotus root increased significantly and reached the maximum of 140.56 J after frying for 2 min. The cohesion of lotus roots did not change significantly after boiling and steaming but increased after deep frying (Table 3). The loss of water during steaming and frying due to high temperatures and the increase of gumminess due to starch gelatinization caused the increased chewiness and cohesiveness of lotus roots.

Amino acids are important flavor substances and can enhance the taste of food. Among 17 amino acids, aspartic acid and glutamic acid provide food with a taste similar to monosodium glutamate (MSG); glycine, serine, threonine and alanine are sweet amino acids; histidine, arginine, valine, methionine, phenylalanine, isoleucine and leucine are bitter amino acids [22]. As for the content of the total amino acids, bitter and sweet amino acids in lotus roots decreased significantly after boiling and steaming, and the longer the cooking time, the more amino acids were lost. The MSG-like amino acids in lotus roots disappeared after boiling and steaming (Table 4). This result was similar to a previous study wherein the amino acid content of cassava leaves decreased significantly after cooking [23]. Frying created a 165℃ high-temperature treatment, which may cause proteolysis, caramelization and Maillard reaction, increased amino acid content and other flavor substances in lotus root; especially the MSG-like amino acids, which increased the most and by up to 53 times that found in fresh lotus roots (Table 4).

Nucleotides are another kind of flavor substance and can interact with amino acids to enhance the flavor of each other. Reports have proven that the nucleotides 5′-GMP and 5′-IMP were considered to be flavor nucleotides, and 5′-GMP has a meat flavor and was a stronger flavor enhancer than MSG [24,25]. Only two nucleotides (5′-CMP and 5′-UMP) were identified in fresh lotus root, and 5′-CMP was the main component of lotus root nucleotides, which possess 71.43% of the total nucleotides (Table 5). This result is similar to a former report [26]. The content of nucleotides in lotus roots increased significantly after cooking; the highest content appeared in steamed material, and the lowest content showed in boiled lotus root. The 5′-CMP is still the main nucleotide in boiled and steamed lotus roots, but it just disappeared after frying. The 5′-UMP content increased dramatically in lotus root after frying, and the 5′-IMP also accumulated in large quantities (Table 5). The nucleotides are synthesized from aspartic acid, glutamic acid, glycine, carbon dioxide and one carbon unit and so on, so the change of amino acid content and high temperature during cooking may cause the change of the nucleotide content in lotus root.

The flavor changes after cooking were closely related to the change in amino acid and nucleotide contents. The radar fingerprint of the electronic tongue showed that, compared with fresh lotus root, three kinds of cooking methods generally decreased five flavor response values with two exceptions: the salty flavor in boiled lotus roots and the umami flavor in fried lotus roots significantly increased (Figure 1). It is easy to understand that the cooking heat destroyed flavor substances and caused the response value of the electronic tongue to decline. Flesh vegetables, including lotus root, accumulate the salty substance as nitrate, polyphenols and flavonoids, but the sweet substances, such as sugars and sweet amino acids, could neutralize their salty flavor. Boiling could dissolve the sugars and destroy the sweet amino acids (Table 4), so the sweet flavor in boiled lotus root declined, and the salty flavor increased. Frying greatly increased the MSG-like amino acid (L-glutamic acid) (Table 4), so the umami flavor in fried lotus root was enhanced dramatically.

PCA is a statistical process that uses orthogonal transformations to convert the observed values of a group of potentially related variables into the values of a group of linearly unrelated variables called principal components [27]. The PCA diagram of five sensors of the electronic tongue showed that the variance contribution rates of the PC1 and the PC2 were 93.47% and 0.19%, respectively, and the cumulative variance contribution rate of the first two PCs was 93.66% (>85%) (Figure 2), indicating that PC1 and PC2 could represent the overall taste feature of samples. The taste of fresh and cooked lotus roots was different and well separated by the electronic tongue in the direction of PC1, and boiled lotus roots possessed the highest variance value at the PC1 axis and closest to the fresh material (Figure 2), implying that the taste of boiled lotus roots was the most natural and representative for cooked lotus roots.

Utilizing an E-nose is a sensitive method for the analysis of volatile flavor substances; it depends on the composition of odor molecules and the concentration of flavor molecules to form the characteristic pattern of smell [27]. The radar map of the electronic nose for lotus roots before and after different types of cooking showed that the signal values of 18 sensors varied with cooking. The S9 signal response value of fresh lotus root was obviously higher than with other sensors (Figure 3), indicating that the dihydrostilbenes compounds were the main characteristic aroma substances of fresh lotus roots. Compared with fresh lotus roots, the signal response value of most cooked lotus roots declined, indicating that cooking could reduce the aroma of lotus roots. The S9 signal response value of lotus root after steaming for 4 min increased significantly (Figure 3), indicating that the content of corresponding volatile flavor compounds (dihydrostilbenes) increased. The S5 signal response value of fried lotus roots was significantly enhanced (Figure 3), showing that the content of biosynthetic compounds, materials produced in the Maillard reaction and baking volatile flavor compounds increased. The smell characteristics of lotus roots after three kinds of cooking were quietly different.

In this study, the PCA diagram of 18 sensors of electronic nose showed that the variance contribution rates of the PC1 and the PC2 were 88.7% and 1.63%, respectively, and the cumulative variance contribution rate of the first two PCs were 90.33% (>85%) (Figure 4), indicating that the odor characteristics of lotus root were well explained. In the direction of PC1 and PC2, the separation effect of 10 samples was good, and there was an obvious separation between fresh, boiled, steamed and fried lotus roots, except for a slight accumulation between boiled and steamed lotus roots (Figure 4). This result was similar to the electronic nose. The odor character of cooked lotus roots was significantly different from the fresh lotus root, and the boiled and fried lotus roots were predominant and similar at the PC1 level, and the steamed and boiled lotus roots at PC2, implying that the smell of boiled lotus roots was the most characteristic after cooking.

Through SPME-GC-MS detection, a total of 58 kinds of volatile flavor compounds were identified in fresh lotus root with a total content of 427.50 μg/Kg. Among them were 19 kinds of alcohol volatile compounds that accounted for 33.40% of the total content; seven kinds of ester volatile compounds, accounting for 23.09% of the total content; and two kinds of olefin volatile compounds, accounting for 17.23% of the total content. Additionally, there are ten kinds of aldehydes volatile compounds, accounting for 14.02% of the total content; eleven kinds of aromatic hydrocarbon volatile compounds, accounting for 6.28% of the total content; three kinds of alkane volatile compounds, accounting for 4.20% of the total content; one kind of hydrazine volatile compounds, accounting for 1.02% of the total content; one kind of ketone volatile compounds, accounting for 0.90% of the total content; and one kind of phenolic volatile compounds, accounting for 0.30% of the total content (Table 6). The volatile compounds of alcohols and esters in fresh lotus root were high, and the aroma characteristics of fresh lotus root were mostly from (E)-2-nonenol, 2-nonen-1-ol, 1-hexanol and dibutyl phthalate. (E)-2-nonenol and 2-nonene-1-ol provided fruity, floral and grass aromas for the overall aroma of lotus root. Dibutyl phthalate, usually the esterified product of corresponding alcohols and carboxylic acids, was thought to contribute to the overall characteristic aroma of lotus root [28].

The flavor of lotus root changed significantly after cooking. After boiling for 4 min, steaming for 4 min and frying for 2 min, 56, 49 and 38 kinds of volatile flavor compounds were identified in lotus roots, and their contents were 277.88 μg/Kg, 365.14 μg/Kg and 4159.41 μg/Kg, respectively. Compared with fresh lotus root, the type and content of volatile compounds in lotus roots decreased after boiling and steaming, the types of alcohol volatile compounds decreased by 11 and 10, respectively, and the content decreased by 73.83% and 74.49%, respectively (Table 6). This result confirmed the report that there was a similar change in the pattern of alcohol volatile compounds in heat-treated mushrooms [29]. After boiling and steaming, the content of benzene derivatives in lotus roots increased significantly, and many kinds of low molecular weight aromatic benzene derivatives, such as toluene, ethylbenzene and xylene, were detected. Xylene isomers were often found in aquatic plants [30]. Toluene and xylene also existed as natural components of plant materials or could be absorbed as environmental pollutants [31].

The content of volatile flavor compounds in lotus root was increased significantly after frying for 2 min (Table 6), especially aldehydes. Hexanal provided fruit and wood flavor; (Z)–2-heptenal provided sweet apricot nut flavor; nonanal had an aroma of wax and fat; trans-2pyrrol 4-decadienal had an irritating flavor, oil flavor, citrus flavor and chicken flavor. The Maillard reaction occurred after high-temperature frying of lotus root and produced a series of heterocyclic compounds, including pyran, pyrazine and pyridine compounds, and ketone volatile flavor compounds also increased significantly, so that the deep-fried lotus roots produced a burnt sweet, pasty, nutty, smoky flavor.

## 5. Conclusions

The texture and flavor of fresh lotus root changed significantly after cooking. After boiling and steaming for 4 min, lotus roots had the best texture properties, rich in flavor amino acids and nucleotides, and produced a large number of benzene derivatives such as toluene, ethylbenzene, xylene and other low molecular weight aromatic benzene derivatives. After deep-frying for 2 min, the texture properties of lotus root were significantly improved, and chewiness was the best among the three cooking methods. The content of amino acids increased significantly, and a large amount of glutamic acid was produced in lotus root after frying. The volatile flavor compounds also increased significantly in lotus roots after frying, especially the aldehydes. The production of pyran, pyrazine and pyridine compounds after frying made the lotus root unique and delicious. The taste and smell of boiled lotus roots were implied to be the most characteristic and representative of cooked lotus roots. These results provide a theoretical basis for research on the texture and flavor of lotus roots before and after cooking and need to be studied further.

## Figures and Tables

**Figure 1 foods-12-02012-f001:**
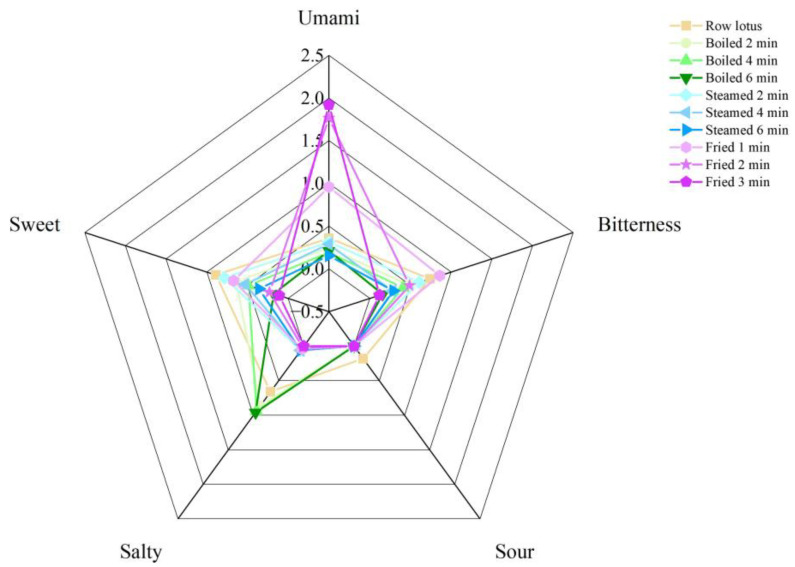
Radar map of electronic tongue of lotus roots before and after cooking.

**Figure 2 foods-12-02012-f002:**
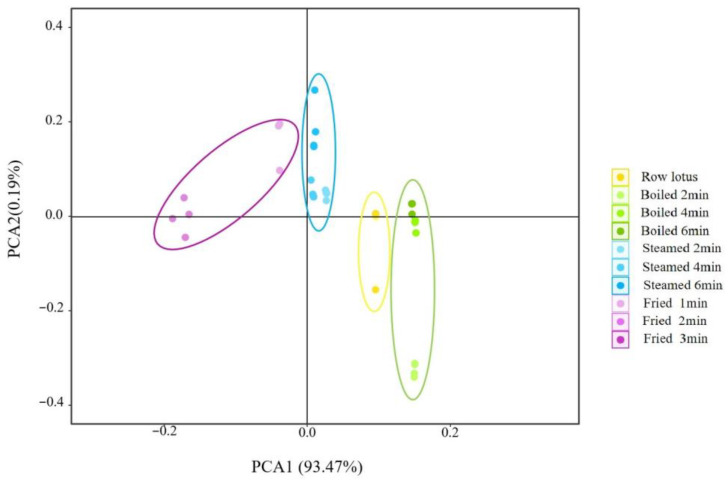
Principal component analysis of electronic tongue of lotus roots before and after cooking.

**Figure 3 foods-12-02012-f003:**
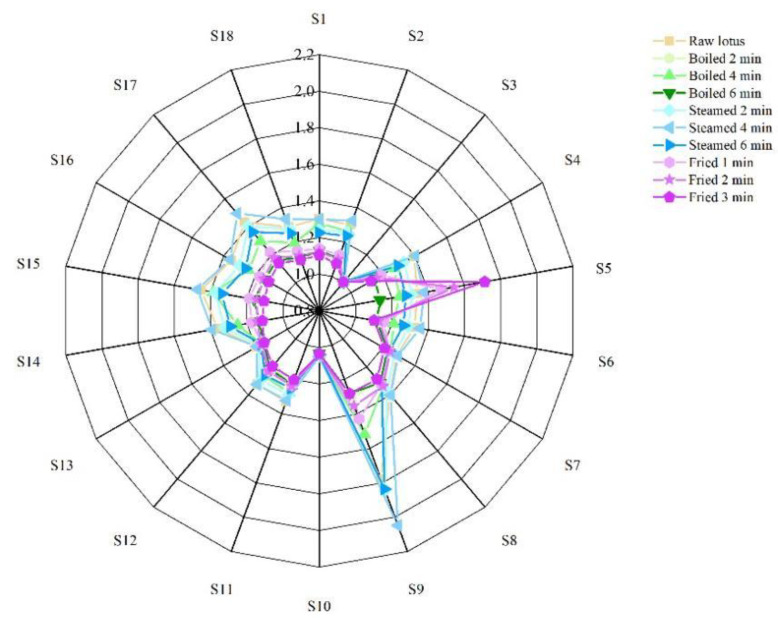
Radar map of electronic nose of lotus roots before and after cooking.

**Figure 4 foods-12-02012-f004:**
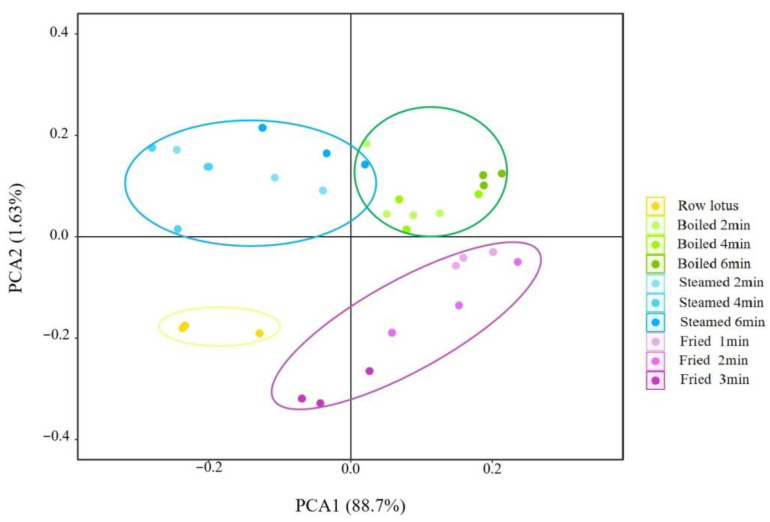
Principal component analysis of electronic nose of lotus roots before and after cooking.

**Table 1 foods-12-02012-t001:** Volatile compounds responded to 18 sensors of the electronic nose.

Sensor No.	Sensing Species
S1	Aromatic compounbds
S2	Oxynitride
S3	Sulfides
S4	Organic acid esters, terpenoids
S5	Biosynthetic compounds, e.g., materials produced in Maillard reaction, baking
S6	Lenthionine
S7	Aliphatic hydrocarbon
S8	Amines
S9	Dihydrostilbenes
S10	Hydrocarbon
S11	TVOC (volatile organic compound)
S12	Sulfide
S13	Ethylene
S14	Volatile gas produced in cooking
S15	Propane compounbds
S16	Isobutane compounbds
S17	combustible gas
S18	Sulfur Compounds

**Table 2 foods-12-02012-t002:** Physicochemical properties of fresh lotus root.

Property	Value
Moisture content (%)	82.99 ± 0.37
Soluble solids (%)	8.03 ± 0.12
Soluble sugar (mg/g)	27.02 ± 1.54
Titratable acid (%)	0.32 ± 0.01

**Table 3 foods-12-02012-t003:** Texture properties of lotus roots before and after cooking.

	Row Lotus	Boiled 2 min	Boiled 4 min	Boiled 6 min	Steamed 2 min	Steamed 4 min	Steamed 6 min	Fried 1 min	Fried 2 min	Fried 3 min
Hardness (N)	243.47 ± 45.32 ^a^	74.57 ± 25.56 ^c^	70.47 ± 4.76 ^c^	64.02 ± 13.67 ^c^	83.57 ± 20.48 ^c^	79.97 ± 16.03 ^c^	73 ± 24.89 ^c^	171.53 ± 71.06 ^b^	226.74 ± 28.73 ^a^	181.53 ± 7.30 ^b^
Springiness	1.96 ± 0.35 ^a^	0.72 ± 0.11 ^d^	0.49 ± 0.04 ^e^	0.56 ± 0.11 ^de^	0.50 ± 0.05 ^e^	0.61 ± 0.16 ^de^	0.65 ± 0.06 ^de^	1.14 ± 0.12 ^c^	1.34 ± 0.15 ^b^	1.31 ± 0.18 ^b^
Gumminess	7.2 ± 0.67 ^c^	6.42 ± 0.61 ^c^	8.33 ± 0.17 ^c^	6.30 ± 0.63 ^c^	10.12 ± 0.49 ^c^	11.35 ± 0.54 ^c^	8.73 ± 0.28 ^c^	61.18 ± 3.55 ^b^	104.48 ± 9.64 ^a^	59.7 ± 5.23 ^b^
Chewiness (J)	5.27 ± 0.34 ^c^	4.75 ± 0.21 ^c^	4.13 ± 0.28 ^c^	3.57 ± 0.19 ^c^	5.17 ± 0.31 ^c^	6.33 ± 0.43 ^c^	5.51 ± 0.35 ^c^	57.56 ± 2.82 ^b^	140.56 ± 9.53 ^a^	79.93 ± 5.83 ^b^
Cohesiveness	0.1 ± 0.01 ^c^	0.09 ± 0.02 ^c^	0.12 ± 0.01 ^c^	0.10 ± 0.01 ^c^	0.12 ± 0.01 ^c^	0.14 ± 0.05 ^c^	0.12 ± 0.02 ^c^	0.33 ± 0.05 ^b^	0.47 ± 0.08 ^a^	0.33 ± 0.05 ^b^

Different subscript letters in the same row for the same item indicate significant differences (*p* < 0.05).

**Table 4 foods-12-02012-t004:** Free amino acid contents of lotus roots before and after cooking.

Free Amino Acids	Content (mg/g Fresh Weight)									
	Row Lotus	Boiled 2 min	Boiled 4 min	Boiled 6 min	Steamed 2 min	Steamed 4 min	Steamed 6 min	Fired 1 min	Fried 2 min	Fried 3 min
L-Aspartic acid	0.29 ± 0.06 ^a^	0.08 ± 0.01 ^b^	ND	ND	0.06 ± 0.02 ^b^	ND	ND	0.02 ± 0.01 ^c^	ND	ND
L-Glutamic acid	ND	ND	ND	ND	ND	ND	ND	6.46 ± 0.21 ^b^	15.25 ± 0.35 ^a^	15.37 ± 0.75 ^a^
L-Serine	0.49 ± 0.11 ^a^	0.33 ± 0.13 ^b^	0.11 ± 0.07 ^c^	0.09 ± 0.02 ^c^	0.06 ± 0.01 ^c^	0.2 ± 0.05 ^b^	0.15 ± 0.08 ^b^	0.13 ± 0.07 ^b^	ND	0.08 ± 0.01 ^c^
L-Histidine	0.11 ± 0.01 ^d^	0.03 ± 0.01 ^e^	0.2 ± 0.05 ^c^	0.14 ± 0.05 ^c^	0.71 ± 0.12 ^b^	0.25 ± 0.12 ^c^	0.05 ± 0.04 ^e^	5.34 ± 1.24 ^a^	4.13 ± 0.96 ^a^	ND
L-Arginine	2.44 ± 0.13 ^b^	1.68 ± 0.08 ^c^	3.21 ± 0.98 ^b^	0.95 ± 0.04 ^e^	2.36 ± 0.11 ^b^	1.8 ± 0.11 ^c^	1.34 ± 0.03 ^d^	2.34 ± 1.23 ^b^	2.97 ± 0.21 ^b^	7.14 ± 1.11 ^a^
L-Threonine	10.82 ± 2.31 ^a^	6.73 ± 1.74 ^b^	4.81 ± 0.15 ^c^	1.26 ± 0.24 ^d^	9.83 ± 2.36 ^a^	7.28 ± 1.45 ^ab^	3.95 ± 0.34 ^c^	1.43 ± 0.21 ^d^	ND	ND
Glycine	0.43 ± 0.14 ^a^	0.48 ± 0.21 ^a^	0.02 ± 0.01 ^c^	0.34 ± 0.05 ^b^	0.42 ± 0.11 ^a^	0.55 ± 0.13 ^a^	0.44 ± 0.11 ^a^	0.26 ± 0.04 ^b^	ND	0.23 ± 0.14 ^b^
L-Tyrosine	0.3 ± 0.11 ^c^	0.21 ± 0.14 ^c^	1.75 ± 0.24 ^a^	0.21 ± 0.12 ^c^	ND	0.6 ± 0.11 ^b^	0.11 ± 0.05 ^c^	ND	ND	0.04 ± 0.01 ^d^
L = Proline	0.09 ± 0.01 ^a^	0.04 ± 0.02 ^b^	ND	ND	ND	ND	ND	ND	ND	ND
L-Alanine	0.22 ± 0.05 ^e^	0.29 ± 0.11 ^e^	0.13 ± 0.08 ^e^	0.24 ± 0.11 ^e^	0.98 ± 0.11 ^d^	0.85 ± 0.11 ^d^	0.94 ± 0.14 ^d^	2.34 ± 0.23 ^b^	4.52 ± 0.45 ^a^	1.23 ± 0.12 ^c^
L-Methionine	0.05 ± 0.02 ^b^	0.01 ± 0.01 ^c^	ND	ND	ND	0.77 ± 0.12 ^a^	ND	ND	ND	ND
L-Valine	0.53 ± 0.11 ^c^	0.41 ± 0.04 ^c^	0.43 ± 0.08 ^c^	0.13 ± 0.03 ^e^	0.23 ± 0.06 ^d^	ND	0.84 ± 0.14 ^b^	3.23 ± 0.73 ^a^	0.31 ± 0.07 ^f^	0.07 ± 0.01 ^i^
L-Tryptophan	0.29 ± 0.14 ^c^	0.21 ± 0.08 ^c^	0.12 ± 0.05 ^d^	0.25 ± 0.17 ^c^	0.54 ± 0.13 ^b^	0.13 ± 0.09 ^cd^	0.27 ± 0.12 ^c^	0.71 ± 0.19 ^a^	ND	0.12 ± 0.05 ^d^
L-Phenylalanine	3.65 ± 0.23 ^b^	2.29 ± 0.12 ^c^	0.28 ± 0.13 ^d^	ND	2.55 ± 0.45 ^c^	0.05 ± 0.01 ^e^	0.13 ± 0.04 ^e^	7.59 ± 0.57 ^a^	0.32 ± 0.12 ^d^	0.11 ± 0.07 ^e^
L-Isoleucine	0.32 ± 0.03 ^a^	0.31 ± 0.04 ^a^	0.11 ± 0.07 ^b^	0.31 ± 0.03 ^a^	0.38 ± 0.04 ^a^	0.04 ± 0.04 ^b^	0.34 ± 0.07 ^a^	0.19 ± 0.08 ^b^	0.07 ± 0.01 ^b^	0.02 ± 0.01 ^c^
L-Lysine	0.38 ± 0.12 ^b^	0.26 ± 0.17 ^b^	ND	0.02 ± 0.01 ^d^	0.32 ± 0.11 ^b^	0.08 ± 0.02 ^c^	ND	0.24 ± 0.11 ^b^	4.52 ± 0.27 ^a^	ND
L-Leucine	0.12 ± 0.09 ^a^	ND	ND	ND	ND	ND	ND	ND	ND	ND
Bitter	7.4 ± 2.36 ^a^	4.91 ± 0.59 ^ab^	4.15 ± 0.97 ^ab^	1.64 ± 0.23 ^c^	6.06 ± 2.13 ^a^	2.79 ± 0.38 ^b^	2.92 ± 044 ^b^	6.47 ± 2.54 ^a^	3.67 ± 0.67 ^b^	0.32 ± 0.07 ^b^
MSG-like	0.29 ± 0.06 ^c^	ND	ND	ND	ND	ND	ND	6.48 ± 0.31 ^b^	15.25 ± 2.13 ^a^	15.37 ± 3.64 ^a^
Sweet	11.94 ± 4.34 ^a^	7.8 ± 2.54 ^a^	4.96 ± 1.38 ^b^	1.86 ± 0.35 ^c^	11.55 ± 5.27 ^a^	8.76 ± 3.78 ^a^	5.33 ± 2.74 ^ab^	4.27 ± 1.39 ^b^	9.04 ± 2.67 ^a^	1.46 ± 0.56 ^c^
Total	20.9 ± 4.69 ^b^	13.36 ± 3.14 ^bc^	11.17 ± 2.67 ^c^	3.94 ± 0.87 ^d^	18.44 ± 4.39 ^b^	12.6 ± 2.95 ^c^	8.56 ± 1.78 ^c^	30.28 ± 5.24 ^a^	32.09 ± 6.74 ^a^	24.41 ± 4.51 ^ab^

Different subscript letters in the same row for the same item indicate significant differences (*p* < 0.05). ND indicate not detected.

**Table 5 foods-12-02012-t005:** 5′-Nucleotides contents of lotus roots before and after cooking.

5′-Nucleotides	Content (mg/g Fresh Weight)									
	Row Lotus	Boiled for 2 min	Boiled for 4 min	Boiled for 6 min	Steamed for 2 min	Steamed for 4 min	Steamed for 6 min	Fried for 1 min	Fried 2 min	Fried 3 min
5′-CMP	0.05 ± 0.07 ^d^	0.31 ± 0.02 ^c^	0.61 ± 0.08 ^b^	0.24 ± 0.12 ^c^	0.93 ± 0.04 ^a^	1.02 ± 0.07 ^a^	0.66 ± 0.17 ^b^	ND	ND	ND
5′-UMP	0.02 ± 0.01 ^c^	ND	0.01 ± 0.01 ^c^	ND	0.02 ± 0.01 ^c^	0.02 ± 0.01 ^c^	0.01 ± 0.01 ^c^	0.25 ± 0.03 ^b^	0.27 ± 0.09 ^b^	0.4 ± 0.02 ^a^
5′-GMP	ND	ND	0.01 ± 0.01 ^b^	ND	0.02 ± 0.01 ^a^	0.02 ± 0.01 ^a^	0.02 ± 0.01 ^a^	ND	ND	ND
5′-IMP	ND	ND	ND	ND	ND	ND	ND	ND	0.58 ± 0.12 ^a^	0.39 ± 0.01 ^b^
5′-AMP	ND	ND	ND	ND	0.01 ± 0.01 ^a^	ND	ND	ND	ND	ND
Total	0.07	0.31	0.63	0.24	0.98	1.06	0.69	0.25	0.85	0.79

Different subscript letters in the same row for the same item indicate significant differences (*p* < 0.05). ND indicate not detected.

**Table 6 foods-12-02012-t006:** Volatile compound contents of lotus roots before and after cooking.

Compound Name (μg/kg)	Row Lotus	Steamed 4 min	Bired 4 min	Fired 2 min
(E)-2-Octen-1-ol	ND	0.97 ± 0.05 ^a^	ND	ND
1-Pentanol	14.54 ± 1.23 ^a^	ND	ND	ND
(Z)-3-Hexen-1-ol	1.2 ± 0.07 ^a^	ND	ND	ND
1-Hexanol	24.06 ± 3.24 ^a^	ND	3.99 ± 0.23 ^b^	ND
5-methyl-1-Hexanol	ND	0.3 ± 0.02 ^b^	0.78 ± 0.11 ^a^	ND
1-Octen-3-ol	ND	ND	ND	32.42 ± 5.26 ^a^
2,4-dimethyl-Cyclohexanol	2.79 ± 0.31 ^b^	ND	ND	33.7 ± 5.14 ^a^
3-Ethyl-4-methylpentan-1-ol	3.05 ± 0.64 ^a^	ND	ND	ND
Benzyl alcohol	4.32 ± 0.97 ^a^	ND	0.58 ± 0.19 ^b^	ND
Furaneol	ND	ND	ND	4.17 ± 0.24 ^a^
2-Octen-1-ol	1.42 ± 0.08 ^c^	2.89 ± 0.23 ^a^	2.11 ± 0.87 ^b^	ND
(E)-2-Octen-1-ol	1.22 ± 0.17 ^c^	ND	9.41 ± 2.19 ^b^	33.7 ± 5.16 ^a^
4-Ethylcyclohexanol	ND	ND	ND	13.4 ± 1.24 ^a^
1-Octanol	12.16 ± 2.14 ^a^	2.89 ± 0.09 ^c^	6.93 ± 1.45 ^b^	ND
Linalool	ND	0.02 ± 0.01 ^a^	ND	ND
(E)-2-Nonen-1-ol	24.81 ± 3.29 ^b^	14.87 ± 1.96 ^c^	ND	133.96 ± 9.48 ^a^
2-Nonen-1-ol	24.81 ± 3.13 ^a^	13.27 ± 1.65 ^b^	6.48 ± 2.13 ^c^	ND
Phenylethyl Alcohol	4.44 ± 0.67 ^a^	ND	ND	ND
(Z)-3-Nonen-1-ol	3.81 ± 0.19 ^a^	ND	ND	ND
1-Nonanol	11.31 ± 2.36 ^a^	ND	ND	ND
5-methyl-2-(1-methylethyl)-Cyclohexanol	3.17 ± 1.69 ^a^	ND	0.74 ± 0.07 ^b^	ND
(E)-2-Decen-1-ol	1.99 ± 0.13 ^b^	ND	4.81 ± 0.94 ^a^	ND
Geraniol	2.99 ± 0.61 ^a^	ND	ND	ND
(Z)-3-Decen-1-ol	1.83 ± 0.24 ^a^	ND	ND	ND
(E)-2-Decenal	0.57 ± 0.07 ^c^	1.06 ± 0.09 ^b^	1.08 ± 0.05 ^b^	87.94 ± 8.91 ^a^
Pentanal	ND	ND	2.37 ± 0.24 ^a^	ND
Hexanal	6.05 ± 2.17 ^c^	3.38 ± 0.87 ^d^	18.91 ± 3.84 ^b^	82.35 ± 9.26 ^a^
Heptanal	ND	0.65 ± 0.08 ^b^	ND	15.58 ± 2.19 ^a^
(Z)-2-Heptenal	ND	ND	6.89 ± 1.67 ^b^	98.95 ± 9.25 ^a^
Benzaldehyde	4.36 ± 0.94 ^a^	0.59 ± 0.07 ^c^	1.58 ± 0.23 ^b^	ND
Octanal	2.79 ± 0.21 ^c^	2.89 ± 0.27 ^c^	9.41 ± 2.19 ^b^	33.7 ± 5.19 ^a^
(E,E)-2,4-Heptadienal	ND	ND	ND	19.54 ± 5.41 ^a^
(E)-2-Nonenal	ND	ND	ND	63.57 ± 8.26 ^a^
(E)-4-Nonenal	ND	ND	ND	25.05 ± 3.45 ^a^
Nonanal	24.81 ± 3.29 ^b^	14.87 ± 1.87 ^c^	3.48 ± 0.98 ^d^	133.96 ± 9.68 ^a^
2-Nonenal	ND	1.01 ± 0.07 ^c^	3.69 ± 0.95 ^b^	19.12 ± 2.36 ^a^
Decanal	1.99 ± 0.13 ^c^	1.03 ± 0.06 ^d^	4.81 ± 0.94 ^b^	12.84 ± 4.29 ^a^
2,4-Nonadienal	ND	ND	ND	15.05 ± 4.57 ^a^
2,4-dimethyl-Benzaldehyde	1.61 ± 0.07 ^a^	0.55 ± 0.05 ^c^	1.03 ± 0.05 ^b^	ND
(Z)-2-Decenal	ND	1.06 ± 0.09 ^b^	1.08 ± 0.05 ^b^	87.94 ± 8.91 ^a^
(Z)-3-Phenylacrylaldehyde	1.24 ± 0.12 ^a^	ND	0.49 ± 0.07 ^b^	ND
2,4-Decadienal	1.69 ± 0.18 ^c^	34.93 ± 5.67 ^b^	1.96 ± 0.08 ^c^	1794.12 ± 56.23 ^a^
(E,E)-2,4-Dodecadienal	ND	ND	ND	0.46 ± 0.02 ^a^
E-2-Undecenal	ND	ND	ND	46.86 ± 5.49 ^a^
Tridecanal	0.58 ± 0.12 ^a^	ND	0.4 ± 0.17 ^a^	ND
13-Methyltetradecanal	14.82 ± 5.26 ^a^	ND	ND	ND
Dodecanal	ND	ND	1.57 ± 0.15 ^a^	ND
Butanoic acid, 2-methyl-, methyl ester	0.59 ± 0.03 ^a^	ND	0.42 ± 0.07 ^b^	ND
Formic acid, heptyl ester	4.09 ± 0.64 ^a^	ND	ND	ND
Formic acid, octyl ester	12.16 ± 2.14 ^a^	ND	6.93 ± 1.45 ^b^	ND
Propanoic acid, 2-methyl-, 3-hydroxy-2,2,4-trimethylpentyl ester	2.65 ± 0.65 ^a^	0.73 ± 0.08 ^b^	2.22 ± 0.15 ^a^	ND
4,7-Methano-1H-inden-6-ol, 3a,4,5,6,7,7a-hexahydro-, acetate	ND	ND	0.49 ± 0.07 ^a^	ND
Dimethyl phthalate	ND	0.36 ± 0.02 ^a^	0.39 ± 0.02 ^a^	ND
5-hexyldihydro-2(3H)-Furanone	0.6 ± 0.05 ^a^	ND	ND	ND
Benzyl Benzoate	5.46 ± 1.24 ^a^	ND	ND	ND
Tetradecanoic acid, 12-methyl-, methyl ester	2.25 ± 0.26 ^a^	ND	ND	ND
Dibutyl phthalate	70.92 ± 4.35 ^a^	0.79 ± 0.12 ^c^	4.93 ± 0.65 ^b^	ND
1,2-Benzenedicarboxylic acid, bis(2-methylpropyl) ester	ND	ND	4.93 ± 0.65 ^a^	ND
1-hydroxy-2-Propanone	ND	ND	ND	56.42 ± 9.21 ^a^
3-Hexanone	1.39 ± 0.24 ^a^	0.65 ± 0.08 ^b^	ND	ND
1-Octen-3-one	ND	ND	ND	7.18 ± 1.26 ^a^
6-methyl-5-Hepten-2-one	ND	ND	1.43 ± 0.21 ^a^	ND
2,5-Dimethylfuran-3,4(2H,5H)-dione	ND	ND	ND	4.17 ± 0.24 ^a^
Acetophenone	1.02 ± 0.07 ^a^	ND	0.64 ± 0.24 ^a^	ND
3-methyl-1,2,4-Cyclopentanetrione	ND	ND	ND	17.19 ± 1.67 ^a^
3-Nonen-2-one	ND	ND	ND	10.1 ± 1.47 ^a^
5-dihydroxy-6-methyl-4H-Pyran-4-one, 2,3-dihydro-3	ND	ND	ND	33.28 ± 5.12 ^a^
trans-3-Nonen-2-one	ND	ND	ND	10.1 ± 1.47 ^a^
2-Dodecanone	ND	ND	1.21 ± 0.14 ^a^	ND
6,10-dimethyl-5,9-Undecadien-2-one	1.66 ± 0.19 ^a^	ND	1.4 ± 011 ^a^	ND
6,10,14-trimethyl-2-Pentadecanone	ND	ND	ND	5.75 ± 1.24 ^a^
2,6,7-trimethyl-Decane	ND	1.44 ± 0.24 ^a^	ND	ND
2,6-dimethyl-Nonane	ND	0.66 ± 0.04 ^b^	0.85 ± 0.09 ^a^	ND
5-methyl-Decane	ND	0.24 ± 0.07 ^b^	0.4 ± 0.02 ^a^	ND
4-methyl-Decane	1.01 ± 0.31 ^b^	1.29 ± 0.14 ^b^	2.75 ± 0.29 ^a^	ND
2-methyl-Decane	ND	ND	0.45 ± 0.05 ^a^	ND
3,7-dimethyl-Nonane	ND	0.51 ± 0.13 ^a^	ND	ND
2,3-Dimethyldecane	ND	ND	0.58 ± 0.19 ^a^	ND
2,6-Dimethyldecane	ND	0.73 ± 0.08 ^a^	ND	ND
5-Ethyldecane	ND	0.47 ± 0.09 ^a^	ND	ND
2-methyl-Undecane	ND	ND	1.03 ± 0.05 ^a^	ND
3-methyl-Undecane	ND	0.47 ± 0.09 ^b^	0.85 ± 0.09 ^a^	ND
4-methyl-Undecane	ND	0.38 ± 0.07 ^a^	ND	ND
5-methyl-Undecane	ND	0.73 ± 0.08 ^a^	ND	ND
4,7-dimethyl-Undecane	ND	ND	2.89 ± 0.98 ^a^	ND
6-methyl-Dodecane	ND	ND	0.42 ± 0.07 ^a^	ND
2,3,5,8-tetramethyl-Decane	ND	0.25 ± 0.01 ^a^	ND	ND
2,9-dimethyl-Decane	ND	ND	0.81 ± 0.17 ^a^	ND
4,6-dimethyl-Dodecane	ND	0.66 ± 0.04 ^a^	ND	ND
7-methyl-Pentadecane	2.3 ± 0.19 ^a^	ND	ND	ND
(1-ethylnonyl)-Benzene	1.19 ± 0.21 ^b^	ND	1.69 ± 0.12 ^a^	ND
Pentadecanal	14.82 ± 5.26 ^a^	ND	ND	ND
(1-pentylheptyl)-Benzene	ND	0.21 ± 0.09 ^a^	ND	ND
(1-butyloctyl)-Benzene	1.4 ± 0.15 ^a^	0.2 ± 0.01 ^b^	ND	ND
(1-propylnonyl)-Benzene	1.05 ± 0.31 ^a^	ND	ND	ND
(1-ethyldecyl)-Benzene	ND	ND	2.32 ± 0.67 ^a^	ND
(1-hexylheptyl)-Benzene	ND	ND	1.13 ± 0.21 ^a^	ND
Toluene	2.12 ± 0.41 ^c^	99.79 ± 6.78 ^b^	112.65 ± 9.67 ^a^	ND
1,3-dimethyl-Benzene	6.3 ± 2.14 ^a^	6.79 ± 2.69 ^a^	ND	ND
Ethylbenzene	6.3 ± 2.14 ^a^	3.37 ± 1.04 ^b^	9.92 ± 3.21 ^a^	ND
p-Xylene	ND	6.79 ± 2.69 ^a^	11.88 ± 2.97 ^a^	ND
1,2,4,5-tetramethyl-Benzene	ND	0.2 ± 0.01 ^a^	ND	ND
(1-butylhexyl)-Benzene	0.75 ± 0.08 ^a^	ND	ND	ND
(1-butylheptyl)-Benzene	1.67 ± 0.19 ^a^	0.1 ± 0.01 ^b^	1.6 ± 0.14 ^a^	ND
(1-propyloctyl)-Benzene	1.4 ± 0.15 ^a^	ND	1.17 ± 0.24 ^a^	ND
(1-methyldecyl)-Benzene	2.34 ± 0.54 ^a^	ND	1.45 ± 0.13 ^b^	ND
(1-pentylheptyl)-Benzene	2.31 ± 0.19 ^a^	ND	1.76 ± 0.21 ^b^	ND
(1-butyloctyl)-Benzene	ND	ND	1.83 ± 0.31 ^a^	ND
(1-propylnonyl)-Benzene	ND	ND	1.28 ± 0.27 ^a^	ND
(1-pentyloctyl)-Benzene	ND	ND	1.13 ± 0.17 ^a^	ND
(1-propyldecyl)-Benzene	ND	ND	0.39 ± 0.04 ^a^	ND
1,3,5-Cycloheptatriene	73.04 ± 6.24 ^b^	49.79 ± 2.15 ^c^	82.65 ± 8.39 ^a^	44.86 ± 3.16 ^c^
3-ethyl-2-methyl-1,3-Hexadiene	ND	ND	ND	52.03 ± 2.16 ^a^
(Z)-5-Undecene	ND	0.4 ± 0.02 ^a^	0.64 ± 0.24 ^a^	ND
(E)-5-Tetradecene	0.63 ± 0.21 ^a^	ND	ND	ND
(Z)-6-Dodecene	ND	ND	ND	11.99 ± 1.36 ^a^
2-Undecenal	ND	0.43 ± 0.05 ^b^	0.23 ± 0.04 ^c^	46.86 ± 5.24 ^a^
Decahydro-1,5,5,8a-tetramethyl-, [1S-(13a48a9R)]-1,2,4-Methenoazulene	ND	0.46 ± 0.05 ^a^	ND	ND
Diepicedrene	ND	0.16 ± 0.01 ^a^	ND	ND
2,4-Ditertbutylphenol	1.45 ± 0.24 ^a^	ND	0.82 ± 0.06 ^b^	ND
Methional	ND	0.62 ± 0.08 ^b^	ND	11.06 ± 2.47 ^a^
Pentanoic acid	ND	ND	ND	32.55 ± 5.39 ^a^
Nonanoic acid	ND	ND	ND	10.11 ± 2.69 ^a^
,methyl-Pyrazine	ND	ND	ND	11.68 ± 5.29 ^a^
2,6-dimethyl-Pyrazine	ND	ND	ND	12.55 ± 2.14 ^a^
2,5-dimethyl-Pyrazine	ND	ND	ND	12.55 ± 2.36 ^a^
(phenylmethyl)-Hydrazine	4.44 ± 0.67 ^a^	ND	ND	ND
,2-pentyl-Pyridine	ND	ND	ND	17.04 ± 2.61 ^a^
Benzothiazole	1.43 ± 0.54 ^a^	ND	ND	ND
1-methyl-Naphthalene	ND	0.31 ± 0.05 ^a^	0.34 ± 0.06 ^a^	ND
1,6-anhydro-D-Glucopyranose	ND	ND	ND	1119.3 ± 54.23 ^a^

Different subscript letters in the same row for the same item indicate significant differences (*p* < 0.05). ND indicate not detected.

## Data Availability

All related data and methods are presented in this paper. Additional inquiries should be addressed to the corresponding author.

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
