# Peer review of "Changes in the Texture and Flavor of Lotus Root after Different Cooking Methods"

_foods, 2023, doi:10.3390/foods12102012_

Round 1

Reviewer 1 Report

The article is quite interesting; experiment was planned properly and generally is well described. The major changes the authors should do is the improvement of the Introduction.

More data can be added in part with description of material - how big were samples obtained for analyses? were there any techological repetitions? it is typical technological experiment and presented results should be average from minimum two technological repetitions.

Author Response

Thank you for your valuable comments and suggestions about our manuscript, Foods-2326286. I really appreciated to your help. The opinions from editor and reviewer were fully considered, and the manuscript was revised as asked.

The detailed corrections are listed below point by point:

Reviewer #1: The article is quite interesting; experiment was planned properly and generally is well described. The major changes the authors should do is the improvement of the Introduction.

Answer: thank you for your suggestion on the introduction, we described lots of basic information in the part of “Result” and “Discussion”, we believe this arrangement could improve the intuitive understanding of this paper for readers. We also consider the length of paper is too long. So we simplified the “introduction”. We also improve the content of introduction.

More data can be added in part with description of material - how big were samples obtained for analyses? were there any techological repetitions? it is typical technological experiment and presented results should be average from minimum two technological repetitions.

Answer: yes, we added the detail information in the revised version.

Finally, the units, dimensions and technical terms were checked again. The reference styles were checked strictly according to the requirement of the Journal of Foods. Please contact me directly if there are some problems with my manuscript. We really appreciated your kind support.

Chunlu Qian

Associated professor

Department of Food Science and Engineering

Yangzhou University

196 Huayang Middle Road, Yangzhou, Jiangsu, P.R. of China

[email protected]

Reviewer 2 Report

The manuscript is within the scope of the Journal. The English language is satisfactory, and readers will understand its content. The authors examined the changes of texture and flavor in raw lotus root and materials after different cooking methods. The results have been appropriately described and discussed. Graphics are visible. The data have been statistically analyzed. However, to maintain the quality and impact of the Journal, it is vital that the authors revise their work according to the following comments:

1. The format/style for the 'Corresponde: author' and 'Abstract' should be checked with the Journal's Instructions for Authors.

2. Introduction: Detailed background of the study should be provided to increase the citations/references. There were only four references/citations provided. This is not appropriate in terms of scientific merit/standard.

3. Section 2.5 should be 'Determination of Free Amino Acids'. Check also Section 2.7.

4. Section 2.10........'an' should be deleted. 

5. The format/style of the 'Sections headings -  should be checked with the Journal's Instructions for Authors. The first letters of the main words should be in Uppercases. (Eg. 2.10. Statistical analyses should be 'Statistical Analyses')

6. All letters (a, b, c, de, etc and ND) should be defined below the Tables 3-6. Tables 3-5 should be laid out in a Landscape format for better visibility of the values. 

7. Table 6 should rather be moved to a 'Supplementary Material' whereas the Table S1 in the Supplementary Material should find its way into the main work. 

8. References: The volume numbers of the Journals in the list of the references should be italicized. The Year should not be placed after the last author name. Please, refer to the Journal's Instructions for Authors for appropriate referencing. 

9. In-text citations should be numbers not the authors names. Please, refer to the Journal's Instructions for Authors for appropriate citations in the text. 

10. The Journal's names in the references list should be abbreviated. Refer to the Journal's Instructions for Authors for appropriate formating. 

11. Sections 3.5 and 3.6 should be properly defined. 'Electronic tongue' and 'Electronic noise' for what, please?

Author Response

Thank you for your valuable comments and suggestions about our manuscript, Foods-2326286. I really appreciated to your help. The opinions from editor and reviewer were fully considered, and the manuscript was revised as asked.

The detailed corrections are listed below point by point:

Reviewer #2: The manuscript is within the scope of the Journal. The English language is satisfactory, and readers will understand its content. The authors examined the changes of texture and flavor in raw lotus root and materials after different cooking methods. The results have been appropriately described and discussed. Graphics are visible. The data have been statistically analyzed. However, to maintain the quality and impact of the Journal, it is vital that the authors revise their work according to the following comments:

  1. The format/style for the 'Corresponde: author' and 'Abstract' should be checked with the Journal's Instructions for Authors.

Answer: yes, we properly revise the format.

  1. Introduction: Detailed background of the study should be provided to increase the citations/references. There were only four references/citations provided. This is not appropriate in terms of scientific merit/standard.

Answer: thank you for your suggestion on the introduction, we described lots of basic information in the part of “Result” and “Discussion”, we believe this arrangement could improve the intuitive understanding of this paper for readers. We also consider the length of paper is too long. So we simplified the “introduction”. We also improve the content of introduction.

  1. Section 2.5 should be 'Determination of Free Amino Acids'. Check also Section 2.7.

Answer: Yes

  1. Section 2.10........'an' should be deleted. 

Answer: Yes

  1. The format/style of the 'Sections headings -  should be checked with the Journal's Instructions for Authors. The first letters of the main words should be in Uppercases. (Eg. 2.10. Statistical analyses should be 'Statistical Analyses')

Answer: Yes

  1. All letters (a, b, c, de, etc and ND) should be defined below the Tables 3-6. Tables 3-5 should be laid out in a Landscape format for better visibility of the values. 

Answer: Yes

  1. Table 6 should rather be moved to a 'Supplementary Material' whereas the Table S1 in the Supplementary Material should find its way into the main work. 

Answer: Yes, we will discuss with the editor about this arrangement.

  1. References: The volume numbers of the Journals in the list of the references should be italicized. The Year should not be placed after the last author name. Please, refer to the Journal's Instructions for Authors for appropriate referencing. 

Answer: Yes, we will check the format again.

  1. In-text citations should be numbers not the authors names. Please, refer to the Journal's Instructions for Authors for appropriate citations in the text. 

Answer: Yes, we will check the format again.

  1. The Journal's names in the references list should be abbreviated. Refer to the Journal's Instructions for Authors for appropriate formating. 

Answer: Yes, we will check the format again.

  1. Sections 3.5 and 3.6 should be properly defined. 'Electronic tongue' and 'Electronic noise' for what, please?

Answer: Yes, we added the detailed information.

Finally, the units, dimensions and technical terms were checked again. The reference styles were checked strictly according to the requirement of the Journal of Foods. Please contact me directly if there are some problems with my manuscript. We really appreciated your kind support.

Chunlu Qian

Associated professor

Department of Food Science and Engineering

Yangzhou University

196 Huayang Middle Road, Yangzhou, Jiangsu, P.R. of China

[email protected]

Reviewer 3 Report

Dear authors,

Expanding of introduction is a priority of this manuscript.

Tables 3-5 need a different perspective and organization. It s very confusing.

There are a few technical errors in the text and tables. Please check and correct it!

Author Response

Thank you for your valuable comments and suggestions about our manuscript, Foods-2326286. I really appreciated to your help. The opinions from editor and reviewer were fully considered, and the manuscript was revised as asked.

The detailed corrections are listed below point by point:

Reviewer #3: Expanding of introduction is a priority of this manuscript.

Answer: thank you for your suggestion on the introduction, we described lots of basic information in the part of “Result” and “Discussion”, we believe this arrangement could improve the intuitive understanding of this paper for readers. We also consider the length of paper is too long. So we simplified the “introduction”. We also improve the content of introduction.

Tables 3-5 need a different perspective and organization. It s very confusing.

Answer: Yes, we will check the format again.

There are a few technical errors in the text and tables. Please check and correct it!

Answer: Yes, we will check it again.

Finally, the units, dimensions and technical terms were checked again. The reference styles were checked strictly according to the requirement of the Journal of Foods. Please contact me directly if there are some problems with my manuscript. We really appreciated your kind support.

Chunlu Qian

Associated professor

Department of Food Science and Engineering

Yangzhou University

196 Huayang Middle Road, Yangzhou, Jiangsu, P.R. of China

[email protected]

Reviewer 4 Report

This is an interesting paper that assesses changes of texture and flavor of lotus root after different cooking methods. It is well organized and the results are interesting. While I have many comments for improving readability and emphasizing key results, the manuscript needs revise according to these comments.

1.     As feedstock for the experiments, the boiling, steam and frying were used to modify changes of texture and flavor of lotus root. One of my initial concerns was whether it would be possible to accurately identify and pinpoint a necessity of suggested coking methodologies. Is it due to the human consumption or industrial processing requirement.  Additionally, the authors are focused on traditional technologies, but do not addressed technologies like freeze-thaw enzyme infusion (https://doi.org/10.1016/j.fbio.2020.100557 , pulsed electric fields (https://doi.org/10.1016/j.lwt.2020.109873) etc. This will make the motivation for this experiment easier to understand in my opinion and introduce the reader better into the topic.

2.     Something else missing from the Introduction is a correlation with papers based on studding of cooking methods on other properties of  lotus root: physicochemical properties and volatile compounds etc.

3.     M&M section 2.3. The authors stated: The texture profile analysis (TPA) used a texture analyzer equipped with TMS-Pro (Food Technology Corporation, USA), in which Texture Lab Pro (texture Index 32 software) is equipped with a cylindrical probe (P/45)….What is the base for choosing such probe? Due to the holes on a lotus root please add information in which part of lotus root slice the cylindrical probe was adopted.

4.     Section 2.2. Please add an explanation of extraction methodology. Which solvent was used for sesame cake extraction? Is it different from well known methods? (https://doi.org/10.1016/j.fbp.2017.09.010)

5.     Conclusion. From reviewer point of view, the cooking methodologies comparison should address its practical approach: industrial processing of lotus root with acces to a final product. The authors do not address this point in a sufficient manner in the Introduction, although they discuss about this in the Results and Discussion chapter. This will make the motivation for this experiment easier to understand in my opinion and introduce the reader better into the topic.

Based on the above points, I would propose a major revision of the manuscript.

Author Response

Thank you for your valuable comments and suggestions about our manuscript, Foods-2326286. I really appreciated to your help. The opinions from editor and reviewer were fully considered, and the manuscript was revised as asked.

The detailed corrections are listed below point by point:

Reviewer #4: This is an interesting paper that assesses changes of texture and flavor of lotus root after different cooking methods. It is well organized and the results are interesting. While I have many comments for improving readability and emphasizing key results, the manuscript needs revise according to these comments.

  1. As feedstock for the experiments, the boiling, steam and frying were used to modify changes of texture and flavor of lotus root. One of my initial concerns was whether it would be possible to accurately identify and pinpoint a necessity of suggested coking methodologies. Is it due to the human consumption or industrial processing requirement.  Additionally, the authors are focused on traditional technologies, but do not addressed technologies like freeze-thaw enzyme infusion (https://doi.org/10.1016/j.fbio.2020.100557 , pulsed electric fields (https://doi.org/10.1016/j.lwt.2020.109873) etc. This will make the motivation for this experiment easier to understand in my opinion and introduce the reader better into the topic.

Answer: Thank you for your suggestion, the boiling, steam and frying are the most common processing method, so we focus on these traditional needs. We also interested in the new processing way like freeze-thaw enzyme infusion and pulsed electric fields, this will be our next work.

  1. Something else missing from the Introduction is a correlation with papers based on studding of cooking methods on other properties of  lotus root: physicochemical properties and volatile compounds etc.

Answer: Thank you for your suggestion, we will improve the introduction section.

  1. M&M section 2.3. The authors stated: The texture profile analysis (TPA) used a texture analyzer equipped with TMS-Pro (Food Technology Corporation, USA), in which Texture Lab Pro (texture Index 32 software) is equipped with a cylindrical probe (P/45)….What is the base for choosing such probe? Due to the holes on a lotus root please add information in which part of lotus root slice the cylindrical probe was adopted.

Answer: we choose the probe according to the texture property of lotus root slice, and also the instruction of equipment. The TPA was conducted on the cross section of lotus root slices. We added the detail in the revised version.

  1. Section 2.2. Please add an explanation of extraction methodology. Which solvent was used for sesame cake extraction? Is it different from well known methods? (https://doi.org/10.1016/j.fbp.2017.09.010)

Answer: Thank you for your suggestion, we will improve the section.

  1. From reviewer point of view, the cooking methodologies comparison should address its practical approach: industrial processing of lotus root with acces to a final product. The authors do not address this point in a sufficient manner in the Introduction, although they discuss about this in the Results and Discussion chapter. This will make the motivation for this experiment easier to understand in my opinion and introduce the reader better into the topic.

Answer: thank you for your suggestion on the introduction, we described lots of basic information in the part of “Result” and “Discussion”, we believe this arrangement could improve the intuitive understanding of this paper for readers. We also consider the length of paper is too long. So we simplified the “introduction”. We also improve the content of introduction.

Finally, the units, dimensions and technical terms were checked again. The reference styles were checked strictly according to the requirement of the Journal of Foods. Please contact me directly if there are some problems with my manuscript. We really appreciated your kind support.

Chunlu Qian

Associated professor

Department of Food Science and Engineering

Yangzhou University

196 Huayang Middle Road, Yangzhou, Jiangsu, P.R. of China

[email protected]

Round 2

Reviewer 2 Report

Minor corrections

1. In-text citations: I understand the the 'in-text citation' should be numbers, and accordingly arranged in the list of references.

2. Format/style for 'section headings/descriptions'. 

Please, check these two comments in reference to the Journal's Instructions for Authors'

Reviewer 4 Report

The authors answered my questions in current manner. Tha manuscript can be accepted.